# A mean-sea-level pressure time series for Trieste, Italy (1841–2018)

Fabio Raicich[1], Renato R. Colucci[1]*

[1]CNR, Institute of Marine Sciences, Trieste, I-34149, Italy
* present address: CNR, Institute of Polar Sciences, Venice-Mestre, I-30172, Italy

5    *Correspondence to*: Fabio Raicich (fabio.raicich@ts.ismar.cnr.it)

**Abstract.** A time series of mean-sea-level pressure was built from observations performed in Trieste from 1 January 1841 to 31 December 2018. Original historical documents provided information on the instruments and on the observation sites. Mercury barometers have been always available. Until 1877 they represented the only instruments in operation, while from 1878 onwards barograph records became available. The time series consists of mean daily values, that were computed from 10   24 hourly data, when possible, or otherwise adjusted to 24-hr means on the basis of climatological daily pressure cycles. The time series was homogenized on the basis of the available metadata, reducing pressure to 0° C and to mean sea level. Basic quality checks were applied, including the comparison with pressure time series from nearby stations. As a result, the data prior to 1865 turned out to be suspect. A mean-sea-level pressure trend of 0.5±0.2 hPa per century was estimated for the 1865–2018 period. The data are available through PANGAEA (https://doi.pangaea.de/10.1594/PANGAEA.926896; Raicich and 15   Colucci, 2021).

## 1 Introduction

Long time series of environmental observations represent key elements for climate studies.

In the last decades, significant efforts have been made in Europe to recover long meteorological time series among which those of the atmospheric pressure. Quality-checked and homogenized time series of daily pressure data were produced for several 20   European cities in the framework of the IMPROVE project, extending from the mid-18[th] century to the 1990's (Camuffo and Jones, 2002). Pressure data for various sites of the Po Plain, in Italy, were recovered by Maugeri et al. (2004). More recently, Cornes et al. (2012a, 2012b) reconstructed over 300-yr long time series for London and Paris, respectively. A comprehensive global inventory of pressure time series that started before 1850 can be found in Brönnimann et al. (2019).

In Trieste regular observations of the atmospheric pressure and the air temperature were reported to start in 1788 (Rossetti, 25   1829), however, only sparse data are available before 1802. Since then the observations have been performed regularly. The earlier observations are only available from local newspapers, while data from 1841 onwards can be found in original manuscripts that are held in the archives of the Institute of Marine Sciences of CNR (CNR-ISMAR) in Trieste (Italy).

This paper focusses on the reconstruction of a daily mean-sea-level pressure time series from 1841 to 2018 using the original observations and the information on the instruments and their heights above mean sea level.

30   In the next section the data used in this work will be described together with their sources and the instruments. The methods used to derive the daily mean-sea-level pressure will be outlined in Sect. 3. Section 4 will include basic information on the data availability. Concluding remarks will be summarized in Sect. 5. Technical details on the barometer corrections and the trend estimates will be presented in Appendices A and B, respectively.

## 2 The data sources

35   ### 2.1 Overview

Pressure observations were initially performed under the responsibility of the local institution committed to the education of future seamen. It operated under different denominations until 1898, namely the Nautical School (*Scuola di Nautica*) until

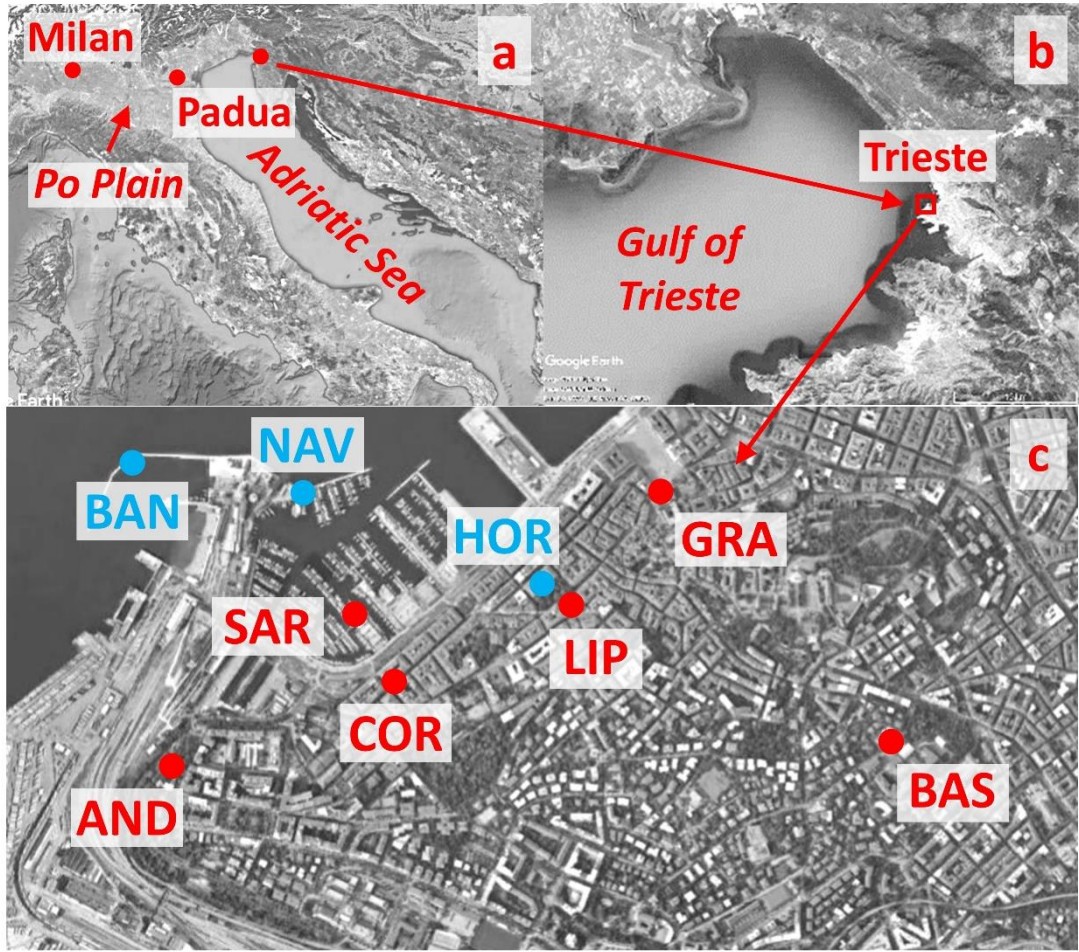

**Figure 1: a) The Adriatic and Po Plain regions; b) the Gulf of Trieste; c) aerial image of Trieste. The observation sites whose data are relevant for this study are displayed in red; other sites cited in the text are shown in light blue. (Images extracted from © Google Earth; © 2020 Landsat/Copernicus, © 2020 CNES/Airbus, © 2020 Digital Globe, © 2020 TerraMetric.)**

1812, the Royal and Nautical Academy (*Accademia Reale e di Nautica*) from 1817 to 1844, the Commercial and Nautical Academy (*Accademia di Commercio e Nautica*) until 1878; in that year the meteorological and the astronomical observatories of the Academy were merged into the Maritime Observatory (*Osservatorio Marittimo*). In 1898 the new Astronomical-Meteorological Observatory (*Osservatorio Astronomico-Meteorologico*), independent of the Academy, was established. In

10   1904 the name was changed back to Maritime Observatory, which was renamed Geophysical Institute (*Istituto Geofisico*) in 1919, and subsequently Thalassographic Institute (*Istituto Talassografico*) in 1941. Finally, in 2002 the latter became part of the newly established Institute of Marine Sciences (*Istituto di Scienze Marine*) of the Italian National Research Council.

Until 1812 the observations were performed in the Public Library situated in the Grand Square (*piazza Grande*, GRA, Fig. 1c). A second-order meteorological station was established in 1817 (Osnaghi, 1886) on the premises of the Royal and Nautical

15   Academy in Leipzig Square (*piazza Lipsia*, LIP, Fig. 1c). In 1841 a new observatory was set up in the same building. In 1868 the meteorological station and service were reorganized. In the first half of that year a new temporary station, located in another room and with new instruments, was operated in parallel with the old one. A fully equipped observatory was finally established on the first floor in July 1868 and the old one was discontinued. Probably in 1868, but certainly before 1884, the observatory became a first-order station (Osnaghi, 1886). In 1870 the station was relocated to the third floor and, finally, in 1876 to a new

20   room on the roof. In 1903 the whole observatory was moved from Leipzig Square, not far from the harbour, to *Villa Basevi* (BAS, Fig. 1c), on a nearby hill, and it was equipped with new instruments. In February 1920 the station was relocated to *Sant'Andrea* (AND, Fig. 1c), again in the low-lying part of the city near the harbour. On 10 June 1944, the building hosting the station partially collapsed due to an aerial bombing and, as a consequence, the barograph and the control barometers were

**Table 1: Summary of the observation sites shown in Fig. 1c, their geographical positions (from Google Earth) and the time intervals when the observations were made.**

| Site | Lat N (°) | Long E (°) | Time interval |
|---|---|---|---|
| GRA – Piazza Grande | 45.6493 | 13.7681 | 8 Jan 1802–30 Sep 1812 |
| LIP – Piazza Lipsia | 45.6468 | 13.7648 | 1 Jan 1819–31 Dec 1902 |
| BAS – Villa Basevi | 45.6442 | 13.7742 | 1 Jan 1903–31 Jan 1920 |
| AND – Sant'Andrea | 45.6435 | 13.7538 | 1 Feb 1920–19 Jul 1944 |
| COR – Via Corsica | 45.6455 | 13.7604 | 19 Jul 1944–22 Aug 1950 |
| AND – Sant'Andrea | 45.6435 | 13.7538 | 22 Aug 1950–31 Dec 2018 |

**Table 2: Summary of the instruments used for the observations. Those indicated with (CB) were also used as control barometers for the barographs.**

| Instrument | Type/sensor | Time interval |
|---|---|---|
| *Barometers* | | |
| Schlosser | siphon | 1 Jan 1841–31 Mar 1865 |
| Hanaczik | adjustable cistern | 1 Jan–31 Dec 1845 |
| Kappeller N. 17 | fixed cistern | 1 Apr 1865–30 Jun 1868 |
| Kappeller N. 710 (CB) | fixed cistern | 1 Jan–31 Mar 1868, 1 Jan 1883–31 Dec 1902 |
| Kappeller N. 1003 (CB) | fixed cistern | 1–22 Apr 1868 |
| Kappeller N. 1004 (CB) | fixed cistern | 23–30 Apr 1868, 1 Jun 1868–31 Dec 1882 |
| Wild-Fuess N. 462 (CB) | adjustable cistern | 1 Jan 1903–31 Dec 2018 |
| SIAP N. 1362 (CB) | adjustable cistern | 1950?–31 Dec 2018 |
| | | |
| *Barographs* | | |
| Kreil | float | 1 Jan 1868–31 Dec 1870, 1 Jan 1878–31 Dec 1902 |
| Sprung-Fuess (Fig 2a) | balance | 1 Jan 1903–31 Mar 1984 |
| Richard N. 86382 | aneroid | Oct 1908–Jan 1979 |
| Fuess 87M N. B5888 (Fig. 2b) | aneroid | 10 May 1956–31 Dec 2018 |
| | | |
| *Digital* | | |
| MICROS BAR | semiconductor | 1994-31 Jan 2000 |
| Vaisälä PTB 201A | capacitive | 1 Feb 2000–31 Dec 2018 |

moved to the University premises in Corsica Street (*via Corsica*, COR, Fig. 1c). In August 1950 the instruments were relocated to the previous site (AND) in the rebuilt institute, and they remained there until the beginning of January 2019, when the station was moved to *Molo* (Pier) *Sartorio* (SAR, Fig. 1c). The positions of the observation sites are summarized in Table 1.

For the sake of completeness, we should mention other meteorological stations where atmospheric pressure is being measured in Trieste, namely: *Molo* (Pier) *Fratelli Bandiera* (BAN, Fig. 1c), managed by the Civil Protection of Friuli Venezia Giulia Region (since 1994); *piazza Hortis* (Hortis Square, present name of Leipzig Square, HOR, in Fig. 1c), managed by the Nautical

Technical Institute and the University of Trieste (since 1993); *Molo* (Jetty) *Lega Navale* (NAV, Fig. 1c), managed by the *Istituto Superiore per la Protezione e la Ricerca Ambientale* (Superior Institute for the Environmental Protection and Research) since 1998.

Data prior to 1841 can only be found in the local newspaper *L'Osservatore Triestino*, which in January 1802 started publishing the observations of pressure, temperature and the 'state of the sky', namely cloudiness and precipitation. The publication of

meteorological data was rather regular until 1808, then increasingly irregular until September 1812 when it ceased; it was resumed in January 1819 and continued regularly except for occasional interruptions.

Original meteorological records and summaries are only available since January 1841; they are held in the archives of CNR-ISMAR. The data from 1 January 1841 to 30 June 1868 were collected in seven handwritten volumes, named *Repertori* (Gallo, 1841–1868). Subsequently, the direct meteorological observations were summarized on monthly data sheets. When automatic

recording instruments became available, hourly data were also tabulated on a monthly basis.

Note that the unavailability of any data before 1802 and of the original observations until 1840 was already stated by Gallo (1846). The second part of this statement is surprising, because he took over the responsibility for the observations on 1 January 1841 but he seemed unaware of the activity performed in the very same place until 31 December 1840. Moreover, he explicitly said that the barometer used in 1841 had also been used previously (Gallo 1841–1868; 1846).

**2.2 The instruments and the observations**

Mercury barometers were available during the whole 1802–2018 period, while barographs were used in 1868–1870 and from 1878 onwards. The station was always equipped with at least one control mercury barometer that was checked against a standard barometer (also named Normal Barometer). Barograph records were always calibrated by means of the observations made with a control barometer. Three types of mercury barometer were used, namely siphon (Gay-Lussac type), with

adjustable cistern (Fortin type) and with fixed cistern (Kappeller type). In order to obtain the atmospheric pressure in standard conditions, that is 0° C temperature and mean sea level (MSL), the observations require appropriate corrections, described in Appendix A.

The original data sheets and summaries generally reported the essential information on the barometers in use, namely model and manufacturer, and height above MSL. Table 2 summarizes the instruments from which the pressure data used in this study

were obtained, as retrieved from the original documents and from review works (Mazelle, 1889; 1905; Polli, 1951–1952; Stravisi, 2006). Unfortunately, the 1802–1840 period lacks of metadata, namely the barometer location, height and temperature, therefore, those data were excluded from further analyses.

Three direct measurements of the atmospheric pressure were made with the control barometer (Table 2) in the morning, in the early afternoon and in the evening, namely at 7, 14 and 22 h from 1 January 1841 to 30 June 1868, then at 7, 14, 21 h until 31

December 1918, at 9, 15, 21 until 31 March 1920, at 8, 13, 18 h until 7 April 1927, and at 8, 14, 19 h approximately until 1980. More recently only one daily observation was performed, usually in the morning.

Until 31 May 1974 pressure was expressed as the height of the mercury column, the units being the Paris line (1 line = 1/12 inch = 1/144 foot, 1 Paris foot = 324.845 mm; Martini, 1883) until 1870 and the millimetre from 1871 onwards. On 1 June 1975 the atmospheric pressure started being expressed in hectopascals.

The 1841–1868 pressure data of the *Repertori* (Gallo, 1841–1868) are generally raw measurements, that is without any corrections and normalizations. According to Gallo (1846) the observations were made at 'true time', that is the local apparent time. On 8 September 1852 the local astronomical observatory began signalling the 'medium Noon', corresponding to 55'3" ahead of Greenwich Mean Time (I.R. Accademia di Commercio e Nautica, 1853). As astronomers and meteorologists were in close contact, perhaps Gallo adopted such a 'medium time' when it became available. Based on the equation of time (Meeus,

1998), the difference between apparent and mean times is between –15 min, on 3 November, and +16 min, on 11 February. Except during extreme events, pressure changes in 15 min are usually small compared to the daily pressure range, therefore, we disregarded that time difference. We also neglected the 5-min difference corresponding to the difference between the station longitude (Table 1) and the 15 °E meridian, where modern standard time is defined.

The barometer temperature is also available making it possible to correct the raw data to 0 °C. The instrumental correction for

the Kappeller N. 17 is available from the *Repertorio* for 1864–1866 (Gallo, 1841–1868) and from Central-Anstalt (1866).

The three daily observations from 1868 to 1902, made with the Kappeller N. 1004 and 710 barometers (Table 2), include uncorrected pressure, barometer temperature and pressure corrected to 0 °C. The instrumental corrections and the constants of normalization to the standard barometer can be found in Central-Anstalt (1884; 1885). In some years, pressure reduced to MSL is also included.

According to the original documents all the direct observations from 1 January 1903 onwards include the correction to account for the barometer temperature and the normalization to a standard barometer, but, unlike the previous data, they are not reduced to MSL. With regard to instrumental corrections and performance, we only know that the Wild-Fuess N. 462 control barometer

(Table 2) was checked by the manufacturer in 1938 and 1952, and that all the control barometers were cross-checked several times (Polli, 1951–1952; Stravisi, 2006).

No information was found about possible instrumental drifts; however, the calibrations and cross-checks have probably allowed to keep them under control.

Besides the three daily observations, pressure was measured every hour on the solstices and equinoxes from 1843 to 1864, and, from 7 to 22 h only, during the whole year 1845. The 1845 data were obtained with a barometer manufactured by Hanaczik (Table 2); the original observations are missing but the daily means were published in a conference communication (Gallo, 1846). Barographs were operated from 1868 to 1870 and from 1878 onwards. The Kreil, Sprung-Fuess and Fuess 87M barographs were used as main instruments, while the Richard barograph worked as a back-up (Table 2).

**3 The 1841–2018 time series**

**3.1 The estimate of daily means**

The time series consists of daily mean pressures reduced to 0 °C and to MSL.

First, the data underwent a preliminary basic quality control in order to recognize and correct evident errors. At that stage we only corrected those errors that could be easily justified, for example, by a writing or printing mistake or missing conversions

to metric units. The latter problem occurred in a limited number of cases from 1871 to 1902, when pressure and temperature were reported in metric units but were still measured with instruments having scales in Paris inches and degrees Réaumur, respectively. Subsequently, pressure changes ($\Delta p$) between adjacent observation times were used to detect suspect values. A visual inspection was carried out when three daily observations were available. By contrast, suspect hourly data were identified when $|\Delta p| > 3$ hPa (the threshold value was chosen arbitrarily). When possible these data were checked in comparison with

the original barograph charts. Erroneous values that could not be corrected were removed from the data set.

Instead of adopting the pressure data reduced to 0 °C reported in the manuscripts for 1868–1902, we recomputed them taking advantage of the availability of observed pressure, barometer temperature, barometer corrections and constants of normalization to the standard barometer.

Thanks to the redundancy of several barometers it was possible to fill almost all the gaps caused by instrumental failures.

Nevertheless, interruptions still exist, particularly in the original hourly record. By contrast, the direct observations at 7, 14 and 21 or 22 h are generally available. Table 3 summarizes the data used to build the 1841–2018 time series.

**Table 3: Chronology of the atmospheric pressure observations composing the 1841-2018 time series. Heights are above mean sea level.**

| Site | Height (m) | Main instrument | Observation times | Time interval |
|---|---|---|---|---|
| LIP – Piazza Lipsia | 14.6 | Schlosser | 7, 14, 22 | 1 Jan 1841–23 Dec 1856 |
| | 23.9 | Schlosser | 7, 14, 22 | 23 Dec 1856–31 Mar 1865 |
| | 23.9 | Kappeller N. 17 | 7, 14, 22 | 1 Apr 1865–30 Jun 1868 |
| | 9.5 | Kappeller N. 1004 | 7, 14, 21 | 1 Jul 1868–1 Jan 1870 |
| | 23.9 | Kappeller N. 1004 | 7, 14, 21 | 1 Jan 1870–1 Sep 1876 |
| | 25.8 | Kappeller N. 1004 | 7, 14, 21 | 1 Sep 1876–31 Dec 1877 |
| | 25.8 | Kreil | hourly | 1 Jan 1878–31 Dec 1902 |
| BAS – Villa Basevi | 67.5 | Sprung-Fuess | hourly | 1 Jan 1903–31 Jan 1920 |
| AND – Sant'Andrea | 12.7 | Sprung-Fuess | hourly | 1 Feb 1920–30 Jun 1920 |
| | 7.8 | Sprung-Fuess | hourly | 1 Jul 1920–19 Jul 1944 |
| COR – Via Corsica | 11.0 | Sprung-Fuess | hourly | 19 Jul 1944–22 Aug 1950 |
| AND – Sant'Andrea | 8.7 | Sprung-Fuess | hourly | 22 Aug 1950–31 Mar 1975 |
| | 8.7 | Fuess 87M N. B5888 | hourly | 1 Apr 1975–31 Jan 2000 |
| | 11.0 | Vaisälä PTB 201A | hourly | 1 Feb 2000–31 Dec 2018 |


We assumed the 'true' daily mean pressure to be obtained by averaging 24 hourly observations, in order to account for the daily cycle. When at least one observation was available a provisional daily mean was computed; if less than 24 data was available, a correction was subsequently applied to adjust it to the 24-hr mean.

Mean corrections were estimated following an approach similar to the one adopted in Raicich and Colucci (2019). For each calendar day (1 January–31 December), climatological values were obtained by averaging hourly (0–23) pressures and mean daily pressures. This was done when all the 24 hourly observations were available, in order to have the full daily cycle represented.

If $h$ is the hour, $d$ the day, $m$ the month, and $y$ the year, let $P_o(h,d,m,y)$ be the observed pressure, $P_c(h,d,m)$ the climatological hourly pressure and $P_{24c}(d,m)$ the climatological daily pressure:

$$P_c(h, d, m) = \sum_{y=y_1}^{y_2}[P_o(h, d, m, y) \cdot w(h, d, m, y)]/\sum_{y=y_1}^{y_2} w(h, d, m, y) \tag{1}$$

$$P_{24c}(d, m) = \sum_{h=0}^{23} P_c(h, d, m)/24, \tag{2}$$

$w$ being a weighting factor, equal to 1 if $P_o$ is available and 0 if it is not; $y_1$ and $y_2$ are the first and last year of the period over which the sums are made. A 91-day running mean is subsequently applied to $P_c$ and $P_{24c}$ in order to smooth out the effect of outliers. The values for 29 February are interpolated using those of 28 February and 1 March.

The MSL pressure at each observation site is characterized by a peculiar daily cycle, mainly related to the air temperature used to estimate the correction to the mean sea level. In fact, the air temperature cycle depends on the exposure of the outdoor thermometer, its distance from buildings and its height above the ground, all of which changed a few times (Table 3). The effect of such issues in historical temperature time series are discussed, for instance, by Cocheo and Camuffo (2002) and Maugeri et al. (2002b). Therefore, we adopted different climatologies to adjust the provisional daily means.

The data for 1841-1870 were adjusted using the 1878-1902 climatology, although the 1868-1870 climatology was also available. We compared the adjustments obtained with both climatologies and we found that the daily differences were mostly less than 0.1 hPa, in absolute value; only in 4% of the cases absolute differences were larger than 0.1 hPa and never larger than 0.2 hPa. Although either period could be adopted, we chose the climatology based on the longer time series, which makes it less sensitive to outliers.

As a consequence, the climatology computed from the 1878–1902 data was used to adjust all the 1841–1902 daily means. Potentially, 25 values of $P_o$ are available for a given $(h,d,m)$ but, because of data gaps, the actual number varies between 21 and 24, except on 26 February (19 data) and 26–27 October (20 data). The 1903–1919 climatology was used to adjust the daily means of the same period; in this case there are 17 or 18 values of $P_o$. The daily means from February 1920 onwards do not require adjustments as 24 hourly observations are always available every day.

The daily value was not estimated only when no data was available, which occurred on: 21 September, 24 September–20 October 1845; 20–25 March and 1 September, 1862; 2 June, 22–23 June, 15–18 July and 30–31 December, 1915. Therefore, 8 days are missing in September 1845, 20 in October 1845, 6 in March 1862, 1 in September 1862, 3 in June 1915, 4 in July 1915 and 2 in December 1915, making a total of 44 missing days out of 65013. The data of September-October 1845 are missing because the barometer temperature is unavailable, and those of 20-25 March 1862 due to the observer's illness (Gallo, 1841-1868). In the other cases the lack of observations occurs for unknown reasons.

The mean daily pressure $P(d,m,y)$ is estimated as:

$$P(d, m, y) = \frac{1}{24}\sum_{h=0}^{23}[P_o(h, d, m, y) \cdot w(h, d, m, y) + P_{adj}(d, m, y) \cdot (1 - w(h, d, m, y))] \tag{3}$$

$$P_{adj}(d, m, y) = P_{24c}(d, m) + \sum_{h=0}^{23}[P_o(h, d, m, y) - P_c(h, d, m)] \cdot w(h, d, m, y)/\sum_{h=0}^{23} w(h, d, m, y) \tag{4}$$

where $w = 1$ if the observation is available and $w = 0$ if it is not. $P_{adj}$ (adjustment) is represented by the daily climatological pressure plus a correction consisting of the mean difference between observed and climatological pressures, computed using

the hourly values available on the relevant day. Clearly, if $P_o$ is available for all hours from 0 to 23, then $P_{adj}$ is not used in Eq. 3 and $P$ is the arithmetic average of the 24 observations.

The error on $P$, namely $\sigma$, is computed from those on the observation, $\sigma_o$, and on the climatologies, $\sigma_{24c}$ and $\sigma_c$, respectively.

$$\sigma(d,m,y) = \frac{1}{24}\left\{\sum_{h=0}^{23}\left[\sigma_o^2(h,d,m,y)\cdot w^2(h,d,m,y) + \sigma_{adj}^2(d,m,y)\cdot\left(1 - w(h,d,m,y)\right)^2\right]\right\}^{\frac{1}{2}} \tag{5}$$

$$\sigma_{adj}^2(d,m,y) = \sigma_{24c}^2(d,m) + \sum_{h=0}^{23}[\sigma_o^2(h,d,m,y) + \sigma_c^2(h,d,m)]\cdot w(h,d,m,y)^2 / [\sum_{h=0}^{23} w(h,d,m,y)]^2 \tag{6}$$

where $\sigma_o$ is the observational error and $\sigma_c$ and $\sigma_{24c}$ the errors on the hourly and daily climatological values, respectively. These errors were assessed semi-empirically as explained in the following. Due to the often uncertain information on the instrumental performances, we did not aim at accurate error estimates but rather at obtaining reasonable representative values. An observation is basically affected by an instrumental error and an environmental error. The instrumental error can be estimated on the basis of the uncertainty on instrumental corrections and normalizations and the nominal reading precision.

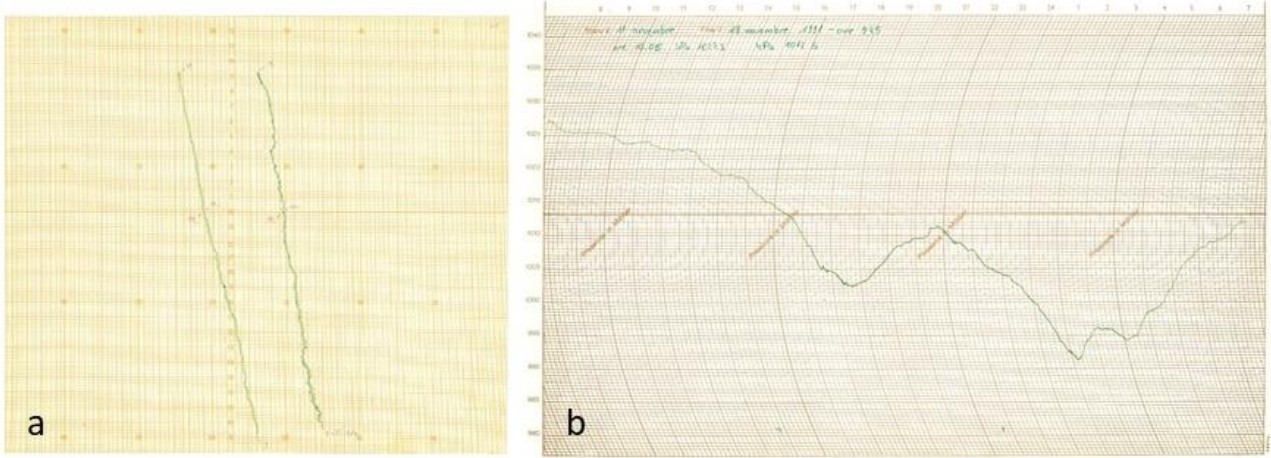

Figure 2: a) Sprung-Fuess barograph chart from 2 February 1954, 10 h, to 4 February 1954, 9 h; b) Fuess 87M barograph chart from 11 November 1991, 10:05 h, to 18 November 1991, 9:45 h.

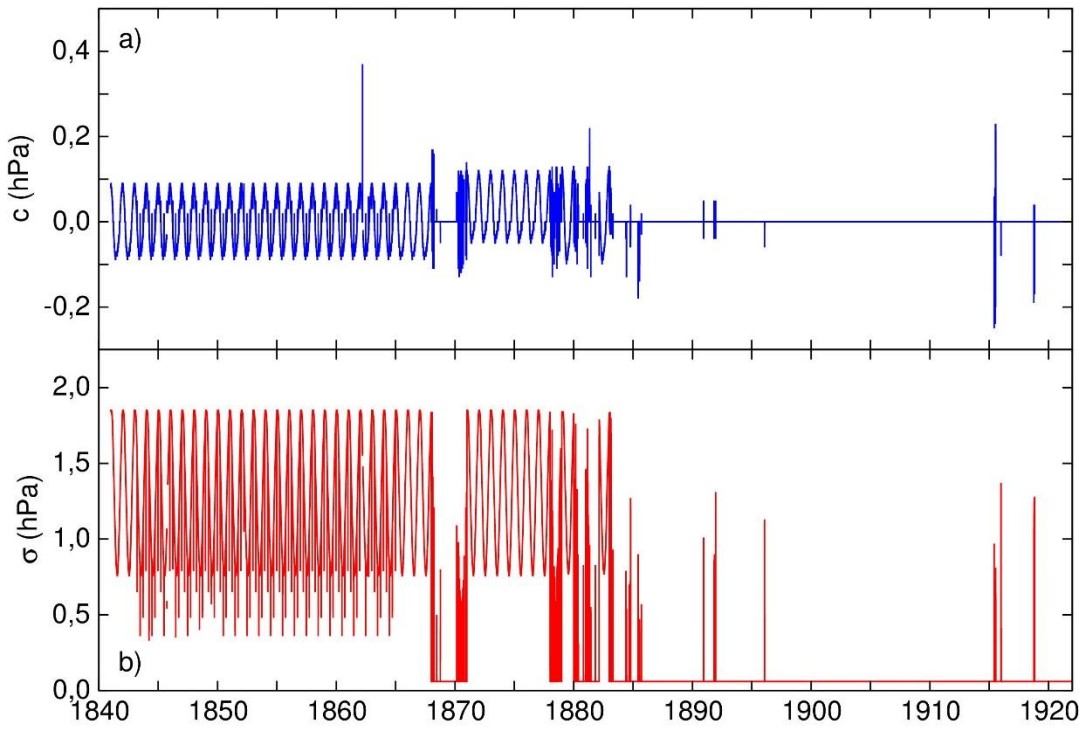

Figure 3: Daily corrections (*c*, panel a) and errors (σ, panel b) from 1841 to 1920 in hPa. From 1920 onwards *c* = 0 hPa and *σ* = 0.06 hPa.

The environmental error is caused by pressure fluctuations occurring at sub-hourly time scales, that may affect the (nominally) instantaneous measurement.

No information was found about errors on the instrumental corrections. With regard to the normalization to a standard barometer, errors were only found for the Kappeller N. 17, 1004 and 710 barometers, namely 0.16, 0.15 and 0.12 hPa, respectively (Central-Anstalt, 1864; 1867; 1868). We assume that a typical normalization error of 0.10–0.15 hPa can be acceptable for the control barometers.

Mercury barometers were provided with verniers, and the nominal reading precisions were 0.1 mm (0.133 hPa) for the Kappeller N. 1004 and 710 barometers (Polli, 1951–1952), and 0.05 mm (0.067 hPa) for the Wild-Fuess N. 462 (Mazelle, 1907; Polli, 1951–1952). The nominal reading precision of barographs can be estimated as half the interval between the chart markings, namely 0.1 mm (0.133 hPa) for the Sprung-Fuess barograph (Fig. 2a) and 0.25 hPa for the Fuess 85M (Fig. 2b). The accuracy of digital instruments is about 0.2 hPa.

The environmental error was estimated as the average hourly pressure range. These were computed using the hourly pressure extremes, available for 2008–2017. The mean hourly pressure range turns out to be 0.13 hPa and on 90% of the days it is lower than 0.24 hPa.

Thus, taking all errors into account, we cautiously assumed the observational error on an individual reading ($\sigma_o$) to be 0.3 hPa, independent of time. The errors on the climatological values, $\sigma_c$ and $\sigma_{24c}$, were obtained as the standard deviations of $P_c$ and $P_{24c}$, respectively (Eq. 1 and 2), to account for the interannual variability of pressure. For both climatologies (1878–1902 and 1903–1919) $\sigma_c$ varies approximately between 4.0 hPa in July and 9.5 hPa in January, while $\sigma_{24c}$ varies between 0.8 and 2.0 hPa, in the same months. These errors are much greater than $\sigma_o$ but they only contribute when the hourly observation is missing. Figure 3 shows the daily corrections (*c*, panel a), to be added to the provisional daily means to adjust them to 24-hr means, and the related errors ($\sigma$, panel b). Only the 1841–1920 period is displayed because, from February 1920 onwards, 24 observations per day are always available, therefore $c = 0$ hPa and $\sigma = 24^{-1/2} \sigma_o = 0.06$ hPa. Two data sampling schemes can be distinguished, namely in 1841–1867 and 1871–1877, when pressure was observed three times per day, and in 1868–1870 and from 1878 onwards, when barograph records were available.

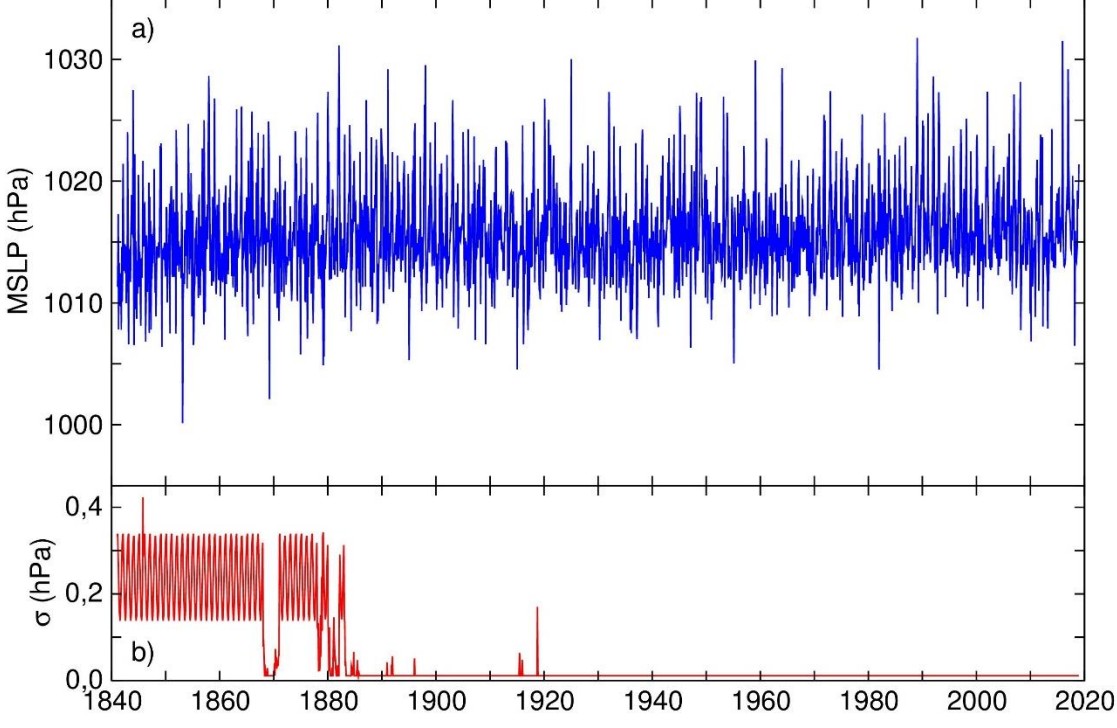

**Figure 4: Monthly mean-sea-level pressure (*MSLP*, panel a) and errors ($\sigma$, panel b) in hPa.**

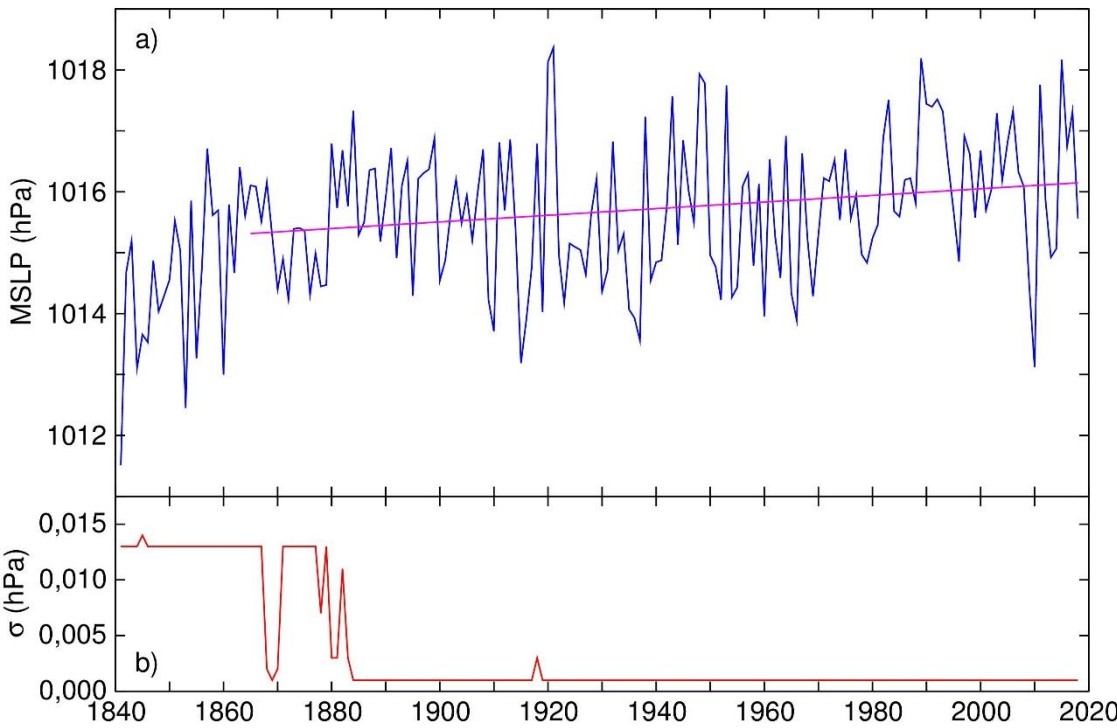

Figure 5: Annual mean-sea-level pressure (*MSLP*, panel a) and errors ($\sigma$, panel b) in hPa. The 1865–2018 linear trend of 0.5±0.2 hPa per century is shown (magenta line).

When three daily observations are available (1841–1867 and 1871–1877) marked annual cycles of both corrections and errors can be seen due to the seasonal variations of the daily cycle. In 1841–1867 corrections normally range between –0.09 and +0.09 hPa with a mean value of –0.01 hPa, while in 1871-1877 the mean correction is +0.03 hPa, ranging between –0.05 and +0.12 hPa. The difference is the result of changing the time of the evening observation from 22 h to 21 h. Note that observing

at 7, 14 and 22 h makes the mean annual bias almost negligible. The spike corresponds to 19 march 1862, when only one observation is available. In 1841–1867 the mean error is 1.30 hPa and it normally ranges between 0.76 and 1.85 hPa. On solstices and equinoxes of 1843-1864 the error drops to 0.4 hPa. In 1868–1870 and from 1878 onwards the magnitudes of corrections and errors reflect the abundance of the observations. In the few cases when less than three observations are available, $|c| > 0.2$ hPa and $\sigma$ attains 1.8 hPa.

The daily means were used to compute monthly (Fig. 4) and annual (Fig. 5) mean pressures and errors. In 1841–1867 and 1871–1877 the monthly error generally varies between 0.15 and 0.35 hPa, and the annual error is than 0.015 hPa. In 1868–1870 and from 1878 onwards monthly errors can be as low as 0.01 hPa (Fig 4b) and annual errors smaller than 0.005 hPa, depending on the amount of observations (Fig 5b).

## 3.2 Comparison with other stations

In order to detect suspect or erroneous data, we compared the daily means, reduced to 0 °C and to MSL, with those of nearby stations, namely the homogenized time series of Padua (1725–1999; Camuffo et al., 2006) and Milan (1763–1998; Maugeri et al., 2002a, b). These time series were selected as they include most of the period of our interest, as only the last 20 years are missing, and because of the relatively short distances from Trieste, namely about 160 km and 350 km, respectively. Another reference was used, namely the daily means of mean-sea-level pressure available on a 2°×2° grid from the 20[th] Century

Reanalysis, version 3 (Compo et al., 2011; Giese et al., 2016; Slivinski et al., 2019), interpolated onto the stations positions. This data set was not used to check the actual pressure values, but rather as a help to detect possible persistent anomalies and drifts in the individual local time series.

Note that anomalous pressure differences can temporarily occur between Trieste and Padua or Milan, situated in the Po Plain, that are not related to the barometer performance. Particularly in autumn and winter the weather in Trieste is often characterized by Bora, a katabatic northeasterly wind with mean hourly speeds that can be higher than 20 m s$^{-1}$ and gusts that can frequently exceed 30 m s$^{-1}$ (Raicich et al., 2013). Gusty wind causes pressure fluctuations that alter the barometer readings (Liu and Darkow, 1989; WMO, 2014), and, therefore, affect the comparison of daily pressure between Trieste and Padua and/or Milan. Typically, strong Bora can last from a couple of days to a week, occasionally up to a couple of weeks (e.g. Raicich et al., 2013).

The daily pressure difference between Trieste and Padua and between Trieste and Milan show anomalous behaviours from 10 March 1844 to 23 May 1846. The 91-day running means, used to smooth day-to-day variability, are displayed in Fig. 6 (only the 1841–1855 period is shown). Seasonal cycles, retained by the running mean, are clearly visible, with higher values in the early months of the year and lower values in the later months. Initially, all the Trieste data reported in the *Repertori* were reduced to 0 °C (Fig. 6, thin lines). However, the amplitudes of the seasonal cycles of the pressure differences appear much larger in March 1844-May 1846. We attributed the anomaly to the fact that, unlike the previous and following observations, the Trieste pressures were written *after* being reduced to 0 °C, and, therefore, were erroneously reduced again. Figure 6 (thick curves) shows that the seasonal cycles appear more coherent when the Trieste data from 10 March 1844 to 23 May 1846 are retained as reported in the *Repertori*.

The anomalous behaviour of the March 1844–May 1846 period was confirmed by comparing the monthly means with those reported in Kreil (1854), Jelinek (1867), Osnaghi (1874) and Mazelle (1886), which were also computed from Gallo's data and explicitly reported to be reduced to 0 °C. Perhaps the pressure measured with the Schlosser barometer was reduced for comparison with the data measured with the Hanaczik, mentioned in Sect. 2.2, which were also reduced to 0 °C and MSL (Gallo, 1846).

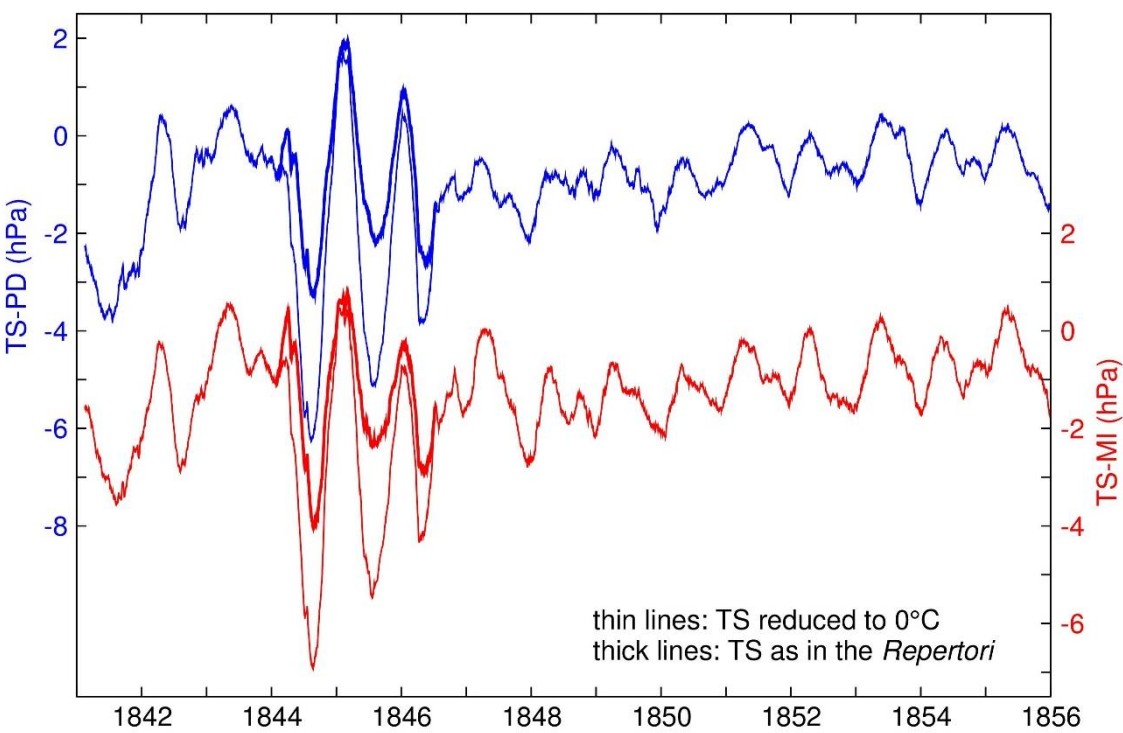

**Figure 6: 91-day running means of daily mean-sea-level pressure differences between Trieste (TS) and Padua (PD) (blue curves, left scale), and between Trieste and Milan (MI) (red curves, right scale). The thin lines correspond to Trieste pressure reduced to 0 °C, the thick lines correspond to Trieste pressure from 10 March 1844 to 23 May 1846 as it appears in the *Repertori*.**

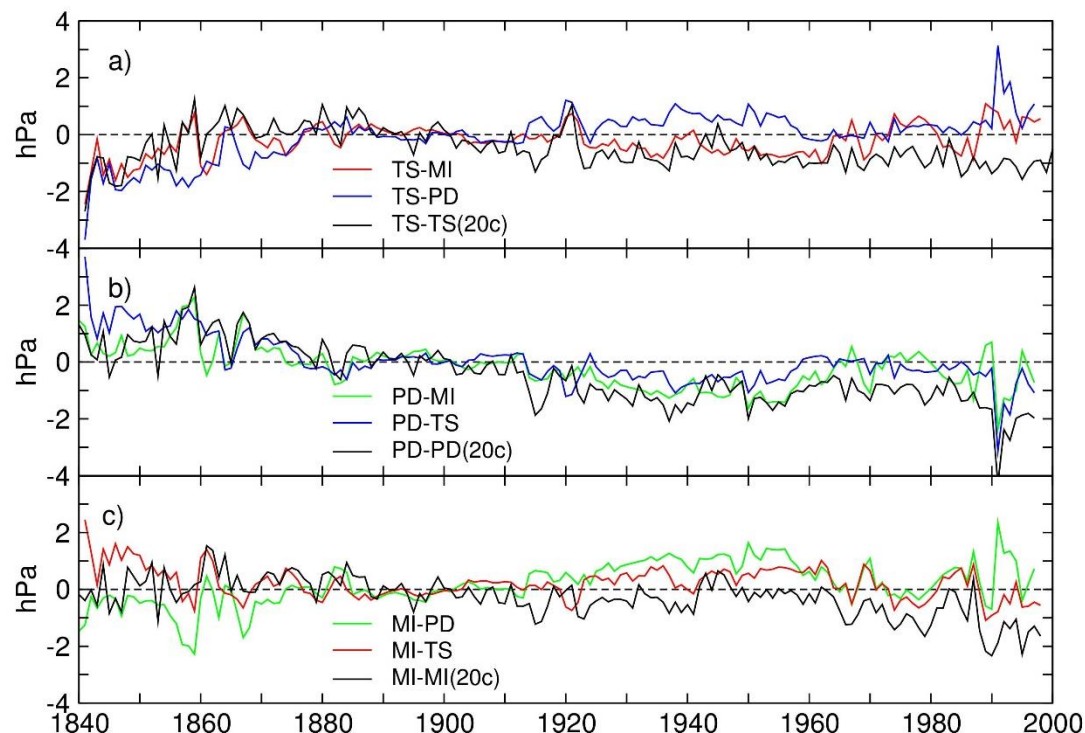

**Figure 7: Comparisons of annual mean pressures (hPa): a) differences involving Trieste (TS); b) differences involving Padua (PD); c) differences involving Milan (MI). Black curves represent the comparisons between each station and the corresponding time series from the 20th Century Reanalysis (20c); the comparison between Trieste and Milan is shown in red, that between Trieste and Padua in blue and that between Padua and Milan in green.**

The annual differences between pressure at Trieste, Padua and Milan are displayed in Fig. 7. In the earlier decades Trieste seems to be lower than expected by at least 1 hPa until the mid-1850's, relative to all the reference time series (Fig. 7a). Another anomaly may occur in the early 1860's but not relative to the reanalysis. The year 1921 also seems anomalous; note that two instrument height changes took place in 1920 (Table 3).

In general, the differences between station data and the respective reanalysis decrease. The average rates from 1861 (after the increase observed at Trieste) onwards are –1.1 hPa per century (Trieste), –1.9 hPa per century (Padua), and –1.2 hPa per century (Milan). This might be related to the reanalysis behaviour in the area of interest, however, for our purpose, it indicates that the Trieste behaviour cannot be attributed to local observations. Another anomaly in the reanalyses is observed in the 1910's, probably connected with lack of good observations during WWI.

Systematic problems are not easy to detect because, when compared to each other, all the time series seem affected by anomalous behaviours. For instance, Padua data seems too high with respect to both Trieste and Milan from 1920 to 1960 (Fig. 7a, 7c) and in the early 1990's (Fig. 7b). Milan itself seems affected by a systematic decrease in the early 1960's, when compared to the reanalysis; Maugeri et al. (2004) pointed out that the Milan station was relocated in 1951 (Fig. 7c).

We can conclude that Trieste pressure of the 1841–1864 period should be considered suspect, while no other major problems can be detected. We recall that in late December 1856 the barometer was moved to a higher position (Table 3) and in April 1865 a new instrument was introduced (Tables 2, 3). Possible reasons of anomalous behaviours of Trieste data are represented by missing instrumental corrections of the Schlosser barometer, used until March 1865, and inaccurate information about instrument heights.

Besides the time series used for the daily data comparisons, monthly time series are available from other stations close to Trieste, namely Zagreb-Grić (Croatia, 1862–2007) and Ljubljana (Slovenia, 1854–2009), both available from the HISTALP data base (Auer et al., 2007), and Udine (Italy, 1803–1855; Meteorologisch Jaarboek, 1871). Overall, these time series allow to corroborate the conclusions drawn from the comparisons with Milan and Padua daily data. In particular, Udine is coherent with Milan and Padua in 1841–1855, thus confirming the anomalous behaviour of Trieste.

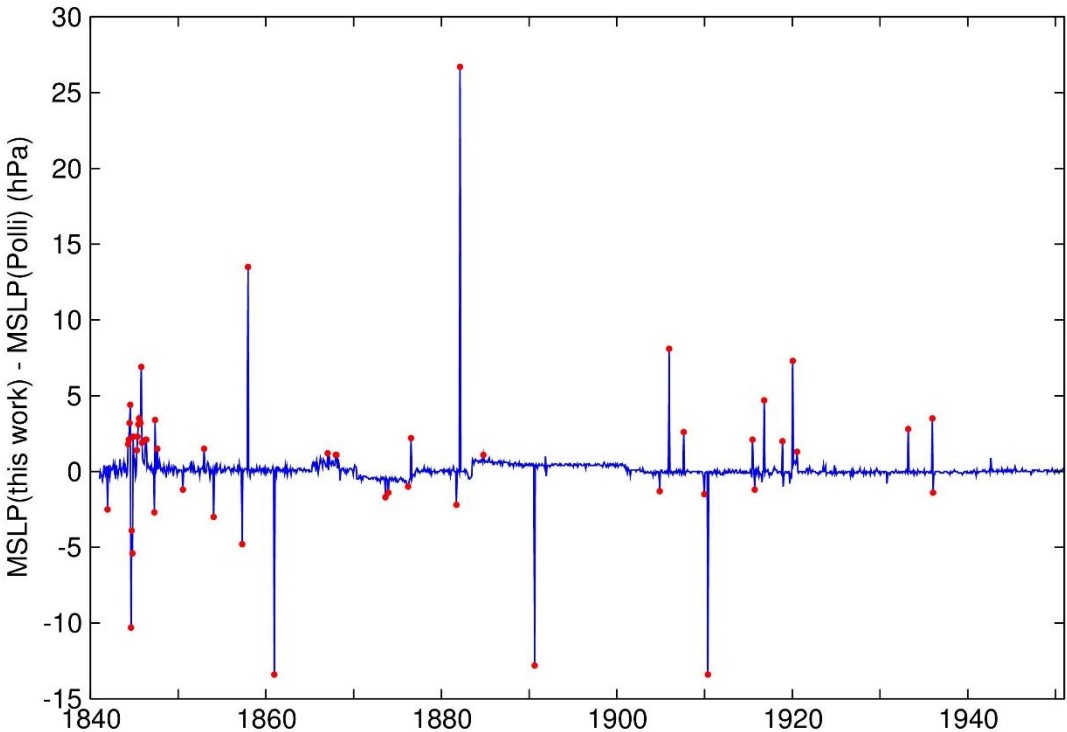

**Figure 8: Differences between monthly pressures from this work and from Polli (1951–1952). Those exceeding 1 mmHg, i.e. about 1.33 hPa, are highlighted by red dots.**

Having discarded the suspect period, Trieste mean-sea-level pressure of 1865–2018 exhibits a trend of 0.5±0.2 hPa per century, significant at $p = 0.02$ (Fig. 5a), after taking into account serial correlation (see details in Appendix B).

### 3.3 Comparison with previous monthly Trieste time series

A time series of mean daily pressure is discussed in this work for the first time, however, in the past monthly time series were already produced by different authors (Kreil, 1854; Jelinek, 1867; Osnaghi, 1874; Mazelle, 1886). Much more recently, Polli
(1951–1952) summarized the monthly pressures, reduced to the MSL, for 1841–1950.

Figure 8 displays the differences between the monthly means from the present work and those in Polli (1951–1952), that were originally reported in millimetres of mercury (mmHg). The red dots highlight the absolute differences exceeding 1 mmHg, corresponding to about 1.33 hPa, which occur in 53 months out of 1320, i.e. 4%.

Discrepancies up to a few tenths of a mmHg are found in most cases, generally due to small differences in calculations or
rounding. The data from March 1844 to May 1846 obviously differ as a consequence of the corrections discussed in Sect. 3.2. The step-like discontinuities in 1865, 1868, 1876 and 1903 correspond to changes of the barometer height (Table 3) and are related to the reduction to the MSL. In fact, for each height, average annual corrections were used until 1902, and constant monthly corrections from 1903 onwards. By contrast, we reduced the individual pressure values before further calculations. At least in four cases, pressure was too high or too low by exactly 10 or 20 mmHg due to misprints. In several cases Polli's
data and those in the original data source do not match. Finally, some mistakes may have occurred in the original calculations.

### 4 Data availability

The 1841-2018 daily mean pressures and the derived monthly and annual mean pressure used in this work are available from PANGAEA as "Mean-sea-level atmospheric pressure from 1841 to 2018 at Trieste, Italy" (https://doi.pangaea.de/10.1594/PANGAEA.926896; Raicich and Colucci, 2021).

## 5 Summary and conclusions

Thanks to the observations performed during 178 years, we built a time series of daily mean atmospheric pressures, reduced to 0 °C and MSL. Data corrections were generally possible thanks to the metadata reported in the original data sources and in contemporary literature. The daily means were computed from 24 hourly values, when possible, or from the available data amount with adjustments based on climatological values. Errors on the daily means were also estimated, that account for the uncertainties on the instruments performances, known from the metadata, and the uncertainties associated to the adjustment to the 24-hour mean, when required.

Although the uncertainties on the individual observations can be rather large, even in the early decades the daily errors are always less than 2 hPa, the monthly errors less than 0.4 hPa and the annual errors less than 0.015 hPa.

A linear trend of 0.5±0.2 hPa per century was estimated for the 1865−2018 period. For comparison with Padua and Milan the trends were computed over the common period 1865-1997, obtaining highly consistent values, namely 0.48±0.25 hPa per century (Trieste), 0.40±0.24 hPa per century (Padua), and 0.52±0.24 hPa per century (Milan).

In conclusion, the whole data set represents a valuable tool to study the pressure variability particularly on monthly and longer time scales.

## Appendix A: Corrections to the barometers readings

We summarize here the main corrections applied to reduce pressure measurements to standard conditions, focussing on the instruments used at Trieste. Jelinek (1876) provided descriptions and discussions of instrument types and methodologies adopted in the 19[th] century in Austria, while more general information can be found in Abbe (1888) and WMO (2014).

### A.1 Gravity

The mercury barometer readings were reduced to the standard gravity of 9.80665 m s$^{-2}$ (WMO, 2014). This is achieved using the relationship:

$$p_g = p \frac{g_{loc}}{g_{std}} \tag{A1}$$

where $p_g$ is pressure reduced to standard gravity, $p$ is the observed pressure, $g_{loc}$ is local gravity acceleration, and $g_{std}$ is standard gravity acceleration. The value $g_{loc} = 9.80568052$ m s$^{-2}$ was obtained in autumn 1991 in the village of Basovizza, about 8 km East from Trieste (Zerbini et al., 1996); as a result, $p_g = 0.9999\ p$, and the correction turns out to be negligible. In the past gravity correction has been disregarded too, because the latitude of Trieste, namely 45.75° N, is very close to 45° N, to which standard gravity used to be referred.

### A.2 Temperature

The barometer readings are affected by the dilatation of the mercury column and of the scale, usually made of brass, caused by temperature variations. The reduction to the standard temperature of 0 °C was computed as:

$$p_0 = p[1 - (\mu - \beta)t] \tag{A2}$$

where $p_0$ is pressure reduced to 0 °C, $p$ is the observed pressure, $\mu = 1.818\ 10^{-4}$ °C$^{-1}$ and $\beta = 0.184\ 10^{-4}$ °C$^{-1}$ are the thermal dilatation coefficients of mercury and brass, respectively, and $t$ is the barometer temperature (°C).

In principle, the Kreil float barograph (Table 2) required a temperature correction, that is unknown, but, in practice, its records were normalized to the direct observations. The Sprung-Fuess barograph was temperature compensated (Mazelle, 1907) as well as the aneroid instruments (Table 2).

## A.3 Barometer corrections

Fixed-cistern barometers require an adjustment to account for the relative change of mercury heights in the tube and in the cistern. Each Kappeller-type instrument used in Trieste was provided with a factory correction (*f*), to be added to the measured value, of the form:

$f = c(p_n - p)$                                                                   (A3)

where *c* is a constant, specific of the individual instrument, $p_n$ is a 'neutral' pressure value and *p* the observed pressure. By contrast, in the Fortin-type barometers the adjustment to a fixed reference level (zero point) is made by means of a screw, therefore, no such correction is needed. Technical details about the various barometer types used in the 19[th] century can be found in contemporary literature, for instance Jelinek (1876), Abbe (1888), as well as in modern articles (e.g. Camuffo et al.,

2010; Brugnara et al., 2015).

Another correction is a constant to be added to normalize the measurement to that of a primary barometer. Before 1919 the reference was the Normal Barometer of the Central Institute for Meteorology und Earth Magnetism in Vienna. By contrast, the primary barometer adopted afterwards is unknown.

Capillarity also alters the reading of a mercury barometer. No specific information was found and we can only assume that its

effect was either already included in the known barometer corrections or it was negligible.

Table A.1 summarizes the values of the barometer and normalization constants that could be found in the literature and in the original documents (Gallo, 1841–1868; Central-Anstalt, 1856, 1866, 1869, 1870, 1874, 1885; Jelinek, 1867; Adria Commission, 1869; Mazelle, 1889). On the basis of the available information we corrected the 1841-1868 data. The data for 1868-1902 were already corrected by the observers, but we re-computed the corrections. The data from 1903 onwards were

corrected by the observers (Sect. 2.2).

**Table A.1: Mercury barometer corrections and constants of normalization to standard barometers.**

| Instrument | Barometer correction (hPa) | Normalization constant (hPa) |
|---|---|---|
| Schlosser | unknown | +0.632 |
| Kappeller N. 17 | 0.074 (1022.58 – p) | –0.162 |
| Kappeller N. 710 | 0.085 (1013.25 – p) | +0.347 |
| Kappeller N. 1004 | 0.061 (1013.25 – p) | +0.453 |
| Wild-Fuess N. 462 | unknown | unknown |
| SIAP N. 1362 | unknown | unknown |


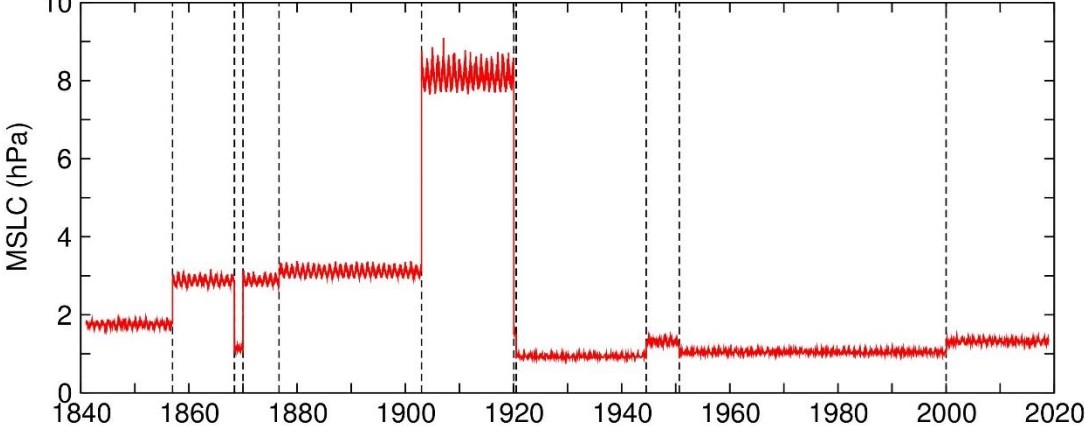

**Figure A1: Daily corrections to reduce pressure to MSL (*MSLC*) in hPa. The dashed vertical lines correspond to the barometer height changes summarized in Table 3.**

## A.4 Reduction to MSL

The equation used to reduce the station pressure to MSL is:

$$p_0 = p_s \, exp\left(\frac{g\,z}{R\,T'_v}\right) \tag{A4}$$

Where $p_0$ is the reduced pressure, $p_s$ is the station pressure, $g$ is the gravity acceleration, $z$ is the barometer height above MSL,

$R = 287.053$ J kg$^{-1}$ K$^{-1}$ is the dry air constant, and $T'_v$ is the adjusted virtual temperature in kelvin. The formula was taken from Stravisi (1988) which includes the details on the involved variables and the calculation, based on Eq. 16 in WMO (1968), having rewritten some variables. T'$_v$ is computed using the observed air temperature and a monthly climatological relative humidity. The adoption of climatological values is justified because the effect of humidity variations on pressure reduction is much smaller than the effect of temperature variations (e.g WMO, 1954).Figure A1 displays the daily time series of the

reductions to MSL, as a function of the barometer heights of Table 3. The seasonal cycle of air temperature causes the high-frequency oscillations.

## Appendix B: Trend with autocorrelation

We estimated the linear trend of the atmospheric pressure time series by linear regression. In order to properly estimate the associated error, the data autocorrelation in time was taken into account. The effect of time autocorrelation is essentially a

lower number of degrees of freedom and, if neglected, the consequent underestimation of the standard error.

To account for autocorrelation, we followed Zervas (2001). A time series consisting of $n$ data points is modelled according to:

$$y_k = bt_k + \rho_1(y_{k-1} - bt_{k-1}) + \varepsilon_k \tag{B1}$$

where $y_k$ is the detrended mean annual pressure ($k$ runs from 1 to $n$), obtained by subtracting the linear trend from the original data, $b$ is the slope of the fitting line, $t_k$ represents time in years, $\rho_1$ is the lag-1 autoregressive coefficient, and $\varepsilon_k$ is the residual.

As a result of autocorrelation, the standard error of the trend increases from $\sigma_b$ to $\sigma_b^{AR}$ (AutoRegressive) by an amount that can be approximated by the square root of the Variance Inflation Factor (VIF):

$$\sigma_b^{AR} = VIF^{1/2}\sigma_b \tag{B2}$$

where

$$VIF = [(1 + \rho_1)/(1 - \rho_1)] \tag{B3}$$

(Maul and Martin, 1993; von Storch and Zwiers, 2001; Wilks, 2006).

## Author contribution

FR retrieved the archived data, prepared the data sets and lead the writing of the paper. RRC was involved in pressure measurement, data collection and processing, and collaborated to the paper writing.

## Competing interests

The authors declare that they have no conflict of interests.

## Acknowledgements

The 20$^{th}$ Century Reanalysis V3 data were provided by the NOAA/OAR/ESRL PSL, Boulder, Colorado, USA, from their web site at https://psl.noaa.gov/.

The authors acknowledge the work done by the previous staff of the Commercial and Nautical Academy, the Maritime

Observatory, the Astronomical-Meteorological Observatory, the Geophysical Institute, the Thalassographic Institute, and the

Institute of Marine Sciences, who managed and performed the observations, and processed and preserved the data used to build the data set. In particular, the authors would like to thank F.A. Immediato, M. Iorio and E. Caterini, in the current staff of the Institute of Marine Sciences of CNR.

The authors thank Prof. Philip Jones and an anonymous reviewer for their comments, that allowed to improve the paper.

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
