# Peer review of "A mean-sea-level pressure time series for Trieste, Italy (1841–2018)"

_Earth System Science Data, 2021_

## Author Response (AR1)

Answers to comments by Reviewer 1 (Philip Jones) (**- RE: ...**)

In the Introduction, there is reference to the long Padua series and that from Milan. The IMPROVE project was one of the first that went back to the original sources and digitized all the sub-daily data from scratch. However, before that long monthly average series have been used for years. The monthly average series for mean sea-level pressure has been used in numerous publications (Jones et al., 1987 and 1999), the latter in the EU ADVICE project. These sources used Polli (1951) update to more recent times. What would be useful in this paper would be to compare your 'new' monthly pressure averages with those from Polli which cover the period from 1841-1950. Polli would have done all the calculations by hand. It would be useful to see how good these were? Did Polli miss something out, or make some mistakes? It would be useful to know how good our forebears were.

**- RE: We did not want to revise and quality-check Polli's monthly means; moreover, the comparison with those data is not straightforward. We just want to point out that the material for 1841-1950 that we examined is the same that Polli had available.**
**We do not know if Polli computed his own means from the original observations. Apparently, he took the original monthly means for granted and, when required, reduced them to the mean sea level. This interpretation should be correct at least until the late 1930's, when Polli became directly involved in the meteorological observations. It is even possible that Polli adopted the data reported in the annual reports for 1884-1911, that were published from 1886 to 1917 (the two references Mazelle, 1907, and Osnaghi, 1886, cited in the paper, belong to that collection), and may themselves include mistakes.**
**Examining the original data sources, the procedure to compute the monthly means seems the following: 1) compute the daily means from three or 24 observations; 2) round the result to the nearest 0.01 / 0.1 Paris line or 0.01 / 0.1 mm, depending on the period; 3) average the rounded daily means to obtain the monthly means. We computed the monthly means from 24-hour (sometimes adjusted) daily means without intermediate rounding.**
**Another source of discrepancies is the reduction to the mean sea level. Polli applied average corrections to the final monthly means. The reduction was based on mean annual corrections until 1902 and mean monthly corrections from 1903 onwards. By contrast, we reduced each pressure value before any further calculations.**
**In general, differences (in absolute value) up to a few tenths of a millimetre (Polli adopted this unit) can occur due to the different computational procedures, and up to about 1 mm in connection with the reduction to the mean sea level. Among the 1841-1950 means reported in Polli (1951), there are 53 months (out of 1320, i.e. 4%) which differ from ours by more than 1 mm (in absolute value). We did not check each case in details, but we found that, at least in four cases, pressure was too high or too low by 10 or 20 mm due to misprints; in several cases Polli's data did not match with those reported in the original manuscripts; finally, in some cases the mistakes were in the original calculations, made by hand.**
**We provided all these details on Polli's data in response to the reviewer's remarks, but, as they represent a sort of revision of Polli's work and because they are not crucial for the time series reconstruction, we think that they should not appear in the paper.**

Focussing just on Italian series from the Po Plain is OK, but there is another long series from Zagreb in Croatia from 1862 (Anon, 1969). This one will be harder for you to find, but it is closer than Milan.

**- RE: The comparisons with other stations and with the 20CR were made to detect possible problems in the Trieste time series, that could not be explained on the basis of the metadata or attributed to erroneous data.
Discrepancies in the comparison beyond normal variability may occur because of problematic data not only in the time series of interest but also in those used for reference. Therefore, to reduce the impact of this problem, we chose to compare Trieste data only with documented homogeneous time series that cover the period of our interest. Within a reasonable distance from Trieste we found two such time series, namely Padua and Milan. Those time series were thoroughly studied in the IMPROVE and ADVICE EU projects. As regards Zagreb, it does not cover the first 21 years of the period of our interest, where Trieste pressure might be too low, and its climate is rather different from Trieste. Moreover, the Zagreb data that we could retrieve were not reduced to the mean sea level (Penzar, I., Juras, J., and, Marki, A., 1992. Long-term meteorological measurements at Zagreb: 1862-1990. Geofizika, 9, suppl., 1-171).
Therefore, we decided not to make the comparison with Zagreb.**

Specific Comments

It is a pity that the earlier observations from 1802 did not provide enough details. Newspaper reports often don't contain enough details. Sometimes, though, with computers and all the data digitized, it is possible to figure out what the numbers mean. Silvio Polli couldn't develop the series earlier than 1841 either, so nothing to be ashamed of.

**- RE: For 1802-1840 we only have the observed pressure at an unknown elevation and the outdoor temperature. Moreover, the data are only available from a local newspaper, that is without additional information about instruments and positions. The values appear reasonable, but, in order to extend the pressure time series back to 1802, a dedicated study would be required. We believe it to be beyond the scope of the present work, focussed on building and describing a time series made from direct observations, rather than reconstructing a time series from any available data. It may be the subject of a future work.**

P3, Line 19, 'relocated to the third floor', and 'to a new room on the roof'. Just better English.

**- RE: (It is page 2) Corrected. (Page 2, line 19, marked-up version)**

I wonder if you're going to look at the long temperature series, but the various site moves might be more important for air temperature than air pressure.

**- RE: Yes, we would like. Unfortunately, at the moment the exact locations and exposures of the outdoor thermometers during the 19$^{th}$ century are sometimes uncertain.**

P4, line 4, change 'told to 'said'.

**- RE: Corrected. (Page 4, line 4, marked-up version)**

Was the same adjustment for temperature used throughout? Sometimes there are Tables giving the adjustments for temperature, elevation and gravity. These Tables may have been slightly altered over the years. I presume you made the adjustments from the original readings. I see later that there are 4 Appendices with the formulae used

**- RE: Yes, as the reviewer said, the relevant adjustments were made from the original readings, using the equations in the four appendices.**

I presume you're confident that the observers all knew what they were doing, even though at some times, it is likely that sometimes junior staff read the instruments. Normally drifts with pressure were spotted quite quickly.

**- RE: We could not but assume that the observations were made by trained staff, and that possible problems could be solved (page 4, lines 40-41 of the submitted manuscript). After all, the station was included in the first-order class network of Austria-Hungary and, later, it was one of the Italian official meteorological stations.**

P5, line 20, better to say 'Thanks to the redundancy of several barometers...'

**- RE: Corrected. (Page 6, line 1, marked-up version)**

P6, with the need to adjust for different observation hours, early observers often measured barometric values every hour for a few years, so they could adjust whatever combination of hours were measured to the true daily mean based on 24 measurements each day. This seems to be based on the 1868-1870 period. Although this is short, you could have smoothed the results?

**- RE: We compared the daily means for 1841-1877, obtained with three observations per day, adjusted using the daily cycles based on the 1868-1870 hourly data and on the 1878-1902 hourly data. In absolute value, the differences are mostly less than 0.1 hPa, therefore, in principle, either period can be adopted. Having also taken into account the remark of another reviewer, we modified the whole paragraph at page 6, lines 6-9 of the submitted manuscript, as follows: "The MSL pressure at each observation site is characterized by a peculiar daily cycle, mainly related to the air temperature used to estimate the correction to the mean sea level. In fact, the air temperature cycle depends on the exposure of the outdoor thermometer, its distance from buildings and its height above the ground, all of which changed a few times (Table 3). The effect of such issues in historical temperature time series are discussed, for instance, by Cocheo and Camuffo (2002) and Maugeri et al. (2002b). Therefore, we adopted different climatologies to adjust the provisional daily means.**
**The data for 1841-1870 were adjusted using the 1878-1902 climatology, although the 1868-1870 climatology was also available. We compared the adjustments obtained with both climatologies and we found that the daily differences were mostly less than 0.1 hPa, in absolute value; only in 4% of the cases absolute differences were larger than 0.1 hPa and never larger than 0.2 hPa. Although either period could be adopted, we chose the climatology based on the longer time series, which makes it less sensitive to outliers." (Page 6, lines 22-31, marked-up version)**
**The relevant additional reference was included:**
**Cocheo, C., and Camuffo, D.: Corrections of systematic errors and data homogenisation in the daily temperature Padova time series (1725-1998). Clim. Change, 53, 77–100, doi: 10.1023/A:1014950306015, 2002. (Page 17, lines 35-36, marked-up version)**

Your adjustments and the equations on p6 all seem good. This is where the check with Polli (1951) would be useful. You can use so much more information and detail than he could! Were there any reasons given for the missing days? Probably WWI was a factor, but also 1845 and 1862. Still 44 out of 65013 is very good.

**- RE: The comparison with Polli's data is discussed above. The reasons for the missing days were explained adding the following text:**
**"The data of September-October 1845 are missing because the barometer temperature is unavailable, and those of 20-25 March 1862 due to the observer's**

**illness (Gallo, 1841-1868). In the other cases the lack of observations occurs for unknown reasons." (Page 6, lines 40-42, marked-up version)**

I presume you're going to adjust the pressure data before 1865? It looks about 1hPa too low. Comparison with neighbours should help – maybe this comes later. The point I'm getting at is that the final few plots indicate issues before 1865.

**- RE: As explained previously, in this paper we wanted to describe a time series built using local data and metadata. Adjustments would require additional information and calculations and could represent the result of a future work.**

With Figure 6, would it not be better comparing with Padua, as it is much close than Milan? Maybe Padua doesn't have the temporal resolution for the period in the mid-1840s.

**- RE: We added the comparison with Padua in Figure 6. The text (page 9, lines 32-38 of the submitted manuscript) was modified as follows:**
**"The daily pressure difference between Trieste and Padua and between Trieste and Milan show anomalous behaviours from 10 March 1844 to 23 May 1846. The 91-day running means, used to smooth day-to-day variability, are displayed in Fig. 6 (only the 1841–1855 period is shown). Seasonal cycles, retained by the running mean, is clearly visible with higher values in the early months of the year and lower values in the later months. Initially, all the Trieste data reported in the Repertori were reduced to 0 °C (Fig. 6, thin lines). However, the amplitudes of the seasonal cycles of the pressure differences appear much larger in March 1844-May 1846. We attributed the anomaly to the fact that, unlike the previous and following observations, the Trieste pressures were written after being reduced to 0 °C, and, therefore, were erroneously reduced again. Figure 6 (thick curves) shows that the seasonal cycles appear more coherent when the Trieste data from 10 March 1844 to 23 May 1846 are retained as reported in the Repertori." (Page 10, line 32 – page 11, line 2, marked-up version)**
**Figure 6 was replaced with a new one. Its legend was modified as:**
**"thin lines – TS reduced to 0 °C**
**thick lines: TS as in the Repertori"**
**(Page 11)**
**and its caption was modified as:**
**"Figure 6: 91-day running means of daily mean-sea-level pressure differences between Trieste (TS) and Padua (PD) (blue curves, left scale), and between Trieste and Milan (MI) (red curves, right scale). The thin lines correspond to Trieste pressure reduced to 0 °C, the thick lines correspond to Trieste pressure from 10 March 1844 to 23 may 1846 as it appears in the Repertori." (Page 11, lines 10-12, marked-up version)**

Figure 7 indicates the problems with Trieste before about 1865. There seems to be an issue with Padua for several years in the early 1990s, and also from the 1920s to the 1960s as you indicate. I am sort of saying that I would like to see a few more conclusions, as opposed to saying that all three Italian series have suspicious periods. So rather than saying Trieste is suspect from 1841-1864, suggest an adjustment. I wouldn't put too much faith in 20CRv3 being correct then.

**- RE: As explained previously, we did not want to adjust pressure at this stage. In the paper (page 11, lines 3-6 of the submitted manuscript) we point out that the anomalies could be related to the fact that during those years the instruments were changed and/or moved a few times. To clarify our conclusion, we moved that sentence to page 12, line 13 – page 13, line 2 of the submitted manuscript). The 20CR time series was also chosen not because it is considered to be the truth, but because data assimilation produces a dynamically coherent product, that, at least, should allow to detect systematic anomalies (page 9, lines 23-24 of the submitted manuscript).**

I hope another reviewer checks whether the data and the format used on the PANGEA site is a good one.

**- RE: A mistake was found in the monthly means file published in PANGAEA. The pressure values were correct, whereas the date column was misaligned in the file retrieved from PANGAEA. The problem arose because the data file that we submitted was re-formatted by PANGAEA staff to comply with their standard. Unfortunately, the mistake was not spotted when the data files were reviewed before publication. The file was corrected.**

Answers to comments by Reviewer 2 (**- RE: ...**)

The manuscript by Raicich and Colucci describes a long daily pressure record for Trieste. The methods are sound and the authors clearly put a lot of effort into recovering and using relevant metadata, which rises significantly the quality of their work. I also appreciate the extra effort in trying to estimate the uncertainty of the pressure values.

On the other hand, I am disappointed by the fact that sub-daily observations are not provided: many important climate products based on pressure data (above all, reanalyses) can hardly make use of daily means. It is also a matter of scientific transparency, as it is not possible to reproduce the results without the raw observations. Therefore I would recommend the authors to publish the hourly/sub-daily observations as well, unless there is a data policy issue that prevents them to do so.

**- RE: Sub-daily raw data at the moment cannot be released for public use.**

As the other reviewer also pointed out, it would be a good idea to compare the results with the Trieste pressure series produced by previous efforts. I actually tried to compare a couple of years of monthly data (1861-1862) with the data published on Austrian yearbooks (https://books.google.ch/books?id=cJA_AAAAcAAJ) and there seem to be a 1-month offset (e.g. the data labeled as "1862-01" is actually for December 1861, please check that!). Even with this error corrected, the differences in monthly means range from +2.6 to -3.6 hPa in those two years (note that yearbook data are not reduced to sea level). Can the methodology alone explain these differences? Or is it perhaps a different station? I assume there might be miscalculation and mistranscription components as well: it could be interesting to know if and how much previously (digitally) available series are affected by them.

**- RE: We thank the reviewer for this remark. The pressure values were correct, while the date column was misaligned in the file retrieved from PANGAEA. The problem arose because the data file that we submitted was re-formatted by PANGAEA staff to comply with their standard. Unfortunately, the mistake was not spotted when the data files were reviewed before publication. Now the file is correct.**

Specific comments

Page 4, Line 26: how certain are you that time was always GMT+00:55? 1842 seems a bit early (not by much, though) for that kind of standardization, which was brought by railways starting from around the 1850s. If a sun clock was used instead, the difference would be up to around 15 minutes depending on the season, which is probably irrelevant anyway (but would be good to know).

**- RE: We thank the reviewer for raising this point.**
**On 8 September, 1852, the local Maritime Government began signalling Noon to enable the seamen in the harbour to reset the ship clocks. The information was provided by the Astronomical Observatory which had been formally established in 1850. However, astronomical observations were being performed since the late 18th century.**
**That time was named the 'medium time of the observatory', corresponding to 55'3" earlier than Greenwich time (I.R. Accademia di Commercio e Nautica, 1853). It is actually unknown whether the 'true time' quoted in Gallo (1846) was the same as the 'medium time'. The availability of astronomical instruments might have enabled Gallo to reset his clock on the basis of the 'medium time' even before 1852, but we cannot be sure.**
**To clarify this point, the text at page 4, lines 26-28 of the submitted manuscript was replaced by:**

"According to Gallo (1846) the observations were made at 'true time', that is the local apparent time. On 8 September 1852 the local astronomical observatory began signalling the 'medium Noon', corresponding to 55'3" ahead of Greenwich Mean Time (I.R. Accademia di Commercio e Nautica, 1853). As astronomers and meteorologists were in close contact, perhaps Gallo adopted such a 'medium time' when it became available. Based on the equation of time (Meeus, 1998), the difference between apparent and mean times is between −15 min, on 3 November, and +16 min, on 11 February. Except during extreme events, pressure changes in 15 min are usually small compared to the daily pressure range, therefore, we disregarded that time difference. We also neglected the 5-min difference corresponding to the difference between the station longitude (Table 1) and the 15 °E meridian, where modern standard time is defined." (Page 4, lines 28-35, marked-up version)
The relevant references were included:
I.R. Accademia di Commercio e Nautica: Prospetto degli studj dell'I.R. Accademia di Commercio e Nautica per l'anno scolastico 1852-1853. Tipografia del Governo, Trieste, 1853, 38 pp. (Page 18, lines 13-14, marked-up version)
Meeus, J.: Astronomical algorithms, Willmann-Bell Inc., Richmond, VA, USA, 1998. (Page 18, line 43, marked-up version)

P4, L40: maybe use a semicolon or a period after "drifts".

- RE: Corrected. (Page 5, line 6, marked-up version)

P5, L13: could you say a few more words about the quality control? What happened to the errors that you could not correct? Would you recommend to perform a finer quality control to the users of your data?

- RE: We thank the reviewer. In fact, the sentence was too concise, therefore, the paragraph at lines 13-16 (submitted version) was re-written as follows:
"First, the data underwent a preliminary basic quality control in order to recognize and correct evident errors. At that stage, we only corrected those errors that could be easily justified, for example, by a writing or printing mistake or missing conversions to metric units. The latter problem occurred in a limited number of cases from 1871 to 1902, when pressure and temperature were reported in metric units but were still measured with instruments having scales in Paris inches and degrees Réaumur, respectively. Subsequently, pressure changes ($\Delta$p) between adjacent observation times were used to detect suspect values. A visual inspection was carried out when three daily observations were available. By contrast, suspect hourly data were identified when $|\Delta p| > 3$ hPa (the threshold value was chosen arbitrarily). When possible these data were checked in comparison with the original barograph charts. Erroneous values that could not be corrected were removed from the data set." (Page 5, lines 20-27, marked-up version)
We built a sort of 'basic' time series, therefore, further quality controls depend on the users' needs.

P6, L6: "Each observation site is characterized by a peculiar mean daily cycle". This is an interesting point, could you explain more in detail what the differences are (amplitude, time shifts?) and what the causes could be?

- RE: The text deals with MSL pressure, which not only involves pressure itself but also indoor temperature, used for the reduction at 0 °C (when required), and outdoor temperature, used for the reduction to the MSL. Actually, air temperature is the main factor affecting the daily cycle, because of different positions and exposures of the thermometers in the various sites. For instance, in the 19[th] century the instrument was first in a window recess, then on the roof, while during most of the 20[th] century it was in a meteorological hut in a garden.

**Having also taken into account the remarks of another reviewer, we modified the whole paragraph at page 6, lines 6-9 of the submitted manuscript, as follows: "The MSL pressure at each observation site is characterized by a peculiar daily cycle, mainly related to the air temperature used to estimate the correction to the mean sea level. In fact, the air temperature cycle depends on the exposure of the outdoor thermometer, its distance from buildings and its height above the ground, all of which changed a few times (Table 3). The effect of such issues in historical temperature time series are discussed, for instance, by Cocheo and Camuffo (2002) and Maugeri et al. (2002b). Therefore, we adopted different climatologies to adjust the provisional daily means.**
**The data for 1841-1870 were adjusted using the 1878-1902 climatology, although the 1868-1870 climatology was also available. We compared the adjustments obtained with both climatologies and we found that the daily differences were mostly less than 0.1 hPa, in absolute value; only in 4% of the cases absolute differences were larger than 0.1 hPa and never larger than 0.2 hPa. Although either period could be adopted, we chose the climatology based on the longer time series, which makes it less sensitive to outliers." (Page 6, lines 22-31, marked-up version)**
**The relevant additional reference was included:**
**Cocheo, C., and Camuffo, D.: Corrections of systematic errors and data homogenisation in the daily temperature Padova time series (1725-1998). Clim. Change, 53, 77–100, doi: 10.1023/A:1014950306015, 2002. (Page 17, lines 35-36, marked-up version)**

P9, Section 3.2: I believe that pressure data from Trieste are assimilated into 20CR starting from 1875, although I am not sure to what extent (see e.g. https://psl.noaa.gov/data/ISPD/). This is relevant for the validation because it means that 20CR might not be independent from your data.
There exists a pressure series from Udine that covers 1803-1855 (https://books.google.ch /books?id=xaxSAAAAcAAJ&pg=PA95&dq=jahrboek+udine&hl=en&sa=X&ved=2ahUKEwiW _p-9nfvAhVDQhoKHRjvCsIQ6AEwAXoECAIQAg# v=onepage&q=jahrboek%20udine&f=false),
perhaps you could try to compare that one too, although the overlapping period is short (but relevant, e.g., for Fig. 6).

**- RE: The comparison with other stations and with the 20CR was made to detect possible systematic problems in the Trieste time series, that could not be explained on the basis of the metadata.**
**Discrepancies in the comparison beyond normal variability may occur because of problematic data not only in the time series of interest but also in those used for reference. Therefore, to reduce the impact of this problem, we chose to compare Trieste data only with documented homogeneous time series that cover the whole period of our interest. Within a reasonable distance from Trieste, we found two such time series, namely Padua and Milan. Those time series were thoroughly studied in the IMPROVE and ADVICE EU projects.**
**The 20CR time series was also chosen because data assimilation produces a dynamically coherent product. The procedure should reduce or eliminate the effect of outliers. Therefore, if Trieste data are too anomalous they should be disregarded in the assimilation process, and the difference between the observations and the 20CR should highlight them.**
**As regards the Udine time series, it just covers 15 years of the period of our interest and we do not know how the data was processed. Those data may represent another source of doubts in the comparison. Figure 6 involves daily data, therefore, Udine monthly data cannot be used; we did not find daily data for Udine.**
**For these reasons, no further comparisons were made.**

Figure 6: I am not 100% sure I understand the problem here: what you label as "not

reduced to 0°C" (red line) was probably actually reduced by the observer and then reduced again by you, is that what you are saying? Wouldn't then be better to not reduce those data? Or is that what you did at the end?

**- RE: It turned out that the anomalous data of March 1844-May 1846 had already been reduced by the observer and that we reduced them again. The key point is that we did it because the observations were reported following the same scheme and style of previous and following data, which did not make us suspicious. The problem arose when comparing the data with Padua and Milan time series. In the publications listed at page 9, line 40, there is enough information to deduce that the data of the original manuscripts (namely the 'Repertori') were already reduced to 0 °C, unlike previous and following data. The comparison with Padua and Milan data is crucial to detect the dates of start and end of the anomaly. The comparison with Padua was introduced taking into account a remark made by another reviewer.**
**We rephrased the text (page 9, lines 32-38, submitted manuscript) to clarify the point.**
**"The daily pressure difference between Trieste and Padua and between Trieste and Milan show anomalous behaviours from 10 March 1844 to 23 May 1846. The 91-day running means, used to smooth day-to-day variability, are displayed in Fig. 6 (only the 1841–1855 period is shown). Seasonal cycles, retained by the running mean, are clearly visible, with higher values in the early months of the year and lower values in the later months. Initially, all the Trieste data reported in the Repertori were reduced to 0 °C (Fig. 6, blue curve). However, the amplitude of the seasonal cycle of the Trieste-Milan pressure difference appears too large in March 1844-May 1846. We attributed this anomaly to the fact that, unlike the previous and following observations, those data were written after being reduced to 0 °C, and, therefore, were erroneously reduced again. Figure 6 (red curve) shows that the seasonal cycle appears more coherent when the Trieste data from 10 March 1844 to 23 May 1846 are retained as reported in the Repertori." (Page 10, line 32 – page 11, line 2, marked-up version)**
**We also modified Fig. 6, by introducing the comparison with Padua, and its legend as:**
**"thin lines – TS reduced to 0 °C**
**thick lines: TS as in the Repertori"**
**(Page 11)**
**Finally, the caption of Fig. 6 was re-written as:**
**"Figure 6: 91-day running means of daily mean-sea-level pressure differences between Trieste (TS) and Padua (PD) (blue curves, left scale), and between Trieste and Milan (MI) (red curves, right scale). The thin lines correspond to Trieste pressure reduced to 0 °C, the thick lines correspond to Trieste pressure from 10 March 1844 to 23 may 1846 as it appears in the Repertori." (Page 11, lines 10-12, marked-up version)**

P11, L35: Trieste is repeated twice.

**- RE: Corrected. (Page 13, line 20, marked-up version)**

P11, L37: I suggest you to rephrase your last sentence in a less negative way, because it gives the impression that daily data before 1880 are not usable, which I do not believe is the case - also considering that you provide an uncertainty estimation attached to the data. To me an error of <2 hPa speaks for rather high quality for early instrumental data.

**- RE: We partly followed the reviewer's suggestion and deleted the relevant sentence. (Page 13, line 22, marked-up version)**

Appendix A.3: please state whether these corrections were applied by you or directly by the observer.

**- RE: The following sentence, summarizing the relevant information, was added at the end of the Appendix:**
**"On the basis of the available information we corrected the 1841-1868 data. The data for 1868-1902 were already corrected by the observers, but we re-computed the corrections. The data from 1903 onwards were corrected by the observers (Sect. 2.2)." (Page 15, lines 5-7, marked-up version)**

Appendix A.4.: please give more details on T'v (data source? is it a climatology?)

**- RE: It is computed from a Fortran code in Stravisi (1988), using the observed air temperature and a climatological relative humidity. Relative humidity has a minor effect on the pressure reduction.**
**To summarize the relevant information, we added the following sentence:**
**"$T'_v$ is computed using the observed air temperature and a monthly climatological relative humidity. The adoption of climatological values is justified because the effect of humidity variations on pressure reduction is much smaller than the effect of temperature variations (e.g. WMO, 1954)." (Page 15, lines 14-16, marked-up version)**
**The relevant reference was included:**
**WMO: Reduction of Atmospheric Pressure (Preliminary Report on Problems Involved), WMO Technical Note No. 7, WMO-No. 36.TP.12, Geneva, Switzerland, available at: https://library.wmo.int/doc_num.php?explnum_id=3443 (last access: 27 January 2021), 1954. (Page 19, lines 38-40, marked-up version)**

**Moreover, 'which is' (page 13, line 14, submitted version) was deleted. (Page 15, line13, marked-up version)**

---

## Author Response (AR2)

**Answers to Report 1 (labelled as Report #2, Referee #1: Philip Jones) – Our replies in bold**

Comment 1
I wasn't asking you to revise and quality check Polli's monthly means from the 1951 paper. You are making many arguments for not doing something that is quite simple. Just compare with the 1841-1950 monthly mean SLP values in the publication.
Some of the points you are making about Polli would be useful, if they could be briefly expressed in your paper. You have spent a lot of time writing almost a page of comments about what Polli did. These comments will be completely forgotten if you don't do anything with them.
The point of a review is not for the authors to give more detail to the reviewer, but to help the authors produce a better paper for readers of the paper in the future.
You have effectively produced a difference series, when you say Polli's data (for 1841-1950) differs by more than 1mm of Hg (presumably this is Hg, as SLP is normally given in hPa) 4% of the time. A histogram of the values would be all that is needed. Then you can explain that the bigger values (10 or 20mm) are clearly typos in Polli's series from 1951. This is all that would be needed. You don't need to do a revision of Polli's 1951 paper, but what you have done allows you to put in the paper the extra work you have done.

**The reason for providing details to the reviewer only is that we interpreted the 'discussion' stage as an opportunity to go beyond simple responses to reviewers' comments and explain our viewpoints more extensively. In a 'traditional' peer-review process we would not have behaved this way. If we were wrong, we apologize.**
**To respond to the reviewer's remark, we included the new Section 3.3 (page 12, lines 7-20, of the marked-up version):**

*3.3 Comparison with previous monthly Trieste time series*
*A time series of mean daily pressure is discussed in this work for the first time, however, in the past monthly time series were already produced by different authors (Kreil, 1854; Jelinek, 1867; Osnaghi, 1874; Mazelle, 1886). Much more recently, Polli (1951-1952) summarized the monthly pressures, reduced to the MSL, for 1841-1950.*
*Figure 8 displays the differences between the monthly means from the present work and those in Polli (1951-1952), that were originally reported in millimetres of mercury (mmHg). The red dots highlight the absolute differences exceeding 1 mmHg, corresponding to about 1.33 hPa, which occur in 53 months out of 1320, i.e. 4%.*
*Discrepancies up to a few tenths of a mmHg are found in most cases, generally due to small differences in calculations or rounding. The data from March 1844 to May 1846 obviously differ as a consequence of the corrections discussed in Sect. 3.2. The step-like discontinuities in 1865, 1868, 1876 and 1903 correspond to changes of the barometer height (Table 3) and are related to the reduction to the MSL. In fact, for each height, average annual corrections were used until 1902, and constant monthly corrections from 1903 onwards. By contrast, we reduced the individual pressure values before further calculations. At least in four cases, pressure was too high or too low by exactly 10 or 20 mmHg due to misprints. In several cases Polli's data and those in the original data source do not match. Finally, some mistakes may have occurred in the original calculations.*

**The new Figure 8 and related caption appear at page 12, lines 1-4, of the marked-up version:**

*Figure 8: Differences between monthly pressures from this work and from Polli (1951-1952). Those exceeding 1 mmHg, i.e. about 1.33 hPa, are highlighted by red dots.*

Comment 2
Zagreb is likely too far away, but it is quite easy to adjust it to mean sea level. It is nearer than Milan though. I just felt that you needed to make more of the comparisons with Padua, as it is much

nearer than Milan. You could just mention that Zagreb is available, and also that an early series might be available for Udine. Udine is available as monthly averages (and is in the list given in Bronnimann et al., 2019), but even at this timescale this might be good enough. You don't have to sort out either Zagreb or Udine, just state that they might be useful in determining the long-term consistency of your Trieste series. You have shown that Polli's data for Trieste only differed from your new series for 4% of the time.

You mention that Padua looks incorrect during the 1990s and I agree with you based on your series and also that from Milan. PD, MI and TS aren't the only series in this part of Italy taking meteorological measurements though. There are likely SLP observations available from the Italian Meteorological Service in Rome, for places such as Verona or Venice Airport. They might be hard to obtain though.

**Please note that the Udine time series from the databank cited in Broennimann et al. (2019) only provide pressure data since 1957, therefore we did not follow the reviewer's suggestion. Another time series was used instead.**
**In Sect. 3.2 the following text was added (page 11, lines 25-29, of the marked-up version):**

*Besides the time series used for the daily data comparisons, monthly time series are available from other stations close to Trieste, namely Zagreb-Grić (Croatia, 1862-2007) and Ljubljana (Slovenia, 1854-2009), both available from the HISTALP data base (Auer et al., 2007), and Udine (Italy, 1803-1855; Meteorologisch Jaarboek, 1871). Overall, these time series allow to corroborate the conclusions drawn from the comparisons with Milan and Padua daily data. In particular, Udine is coherent with Milan and Padua in 1841-1855, thus confirming the anomalous behaviour of Trieste.*

**New relevant references:**

**(Page 16, lines 21-25, of the marked-up version):**
*Auer, I., Böhm, R., Jurkovic, A., Lipa, W., Orlik, A., Potzmann, R., Schöner, W., Ungersböck, M., Matulla, C., Briffa, K., Jones, P., Efthymiadis, D., Brunetti, M., Nanni, T., Maugeri, M., Mercalli, L., Mestre, O., Moisselin, J.-M., Begert, M., Müller-Westermeier, G., Kveton, V., Bochnicek, O., Stastny, P., Lapin, M., Szalai, S., Szentimrey, T., Cegnar, T., Dolinar, M., Gajic-Capka, M., Zaninovic, K., Majstorovic, Z., and Nieplova, E.: HISTALP—Historical instrumental climatological surface time series of the greater Alpine region, Int. J. Climatol., 27, 17–46, doi: 10.1002/joc.1377, 2007.*

**(Page 18, lines 32-34, of the marked-up version):**
*Meteorologisch Jaarboek: Meteorologisch Jaarboek voor 1870, Koninklijk Nederlandsch Meteorologisch Instituut, Utrecht, The Netherlands, available at: https://opacplus.bsb-muenchen.de/Vta2/bsb11035259/bsb:43925481871 (last access 2 June 2021), 1871.*

Comment 3
It is worth mentioning that you might work on the earlier data – in the paper, not just to me.
I agree that developing a long temperature series would be much more work, but it would be a useful piece of work. There are likely more long-term temperature series in the region than there are SLP ones.

**As explained above, the answers to the reviewer were also meant to be a 'discussion'. We do not agree with the reviewer on both points.**
**1) He suggests to mention a work that was not done (although it might be in the future). As explained at page 4, lines 16-17, we did not do it due to the lack of metadata and auxiliary information.**

**2) Secondly, reconstructing a temperature time series is completely out of the scope of the paper. We also believe that a pressure time series is more valuable exactly because there are fewer of them compared to temperature, as the reviewer said.**
**For these reasons, the text was not changed.**

Comment 4
Thanks for the revisions about the diurnal cycles, and also possible ideas about the missing observations.

**No answer.**

Comment 5
There is still the issue of the long-term homogeneity of the record before 1865. You're presenting a series that clearly has a problem. You have added some text, but it would be useful to mention that your series must have an issue before 1865 when compared to Padua and Milan.

**The whole section 3.2 is devoted to comparisons with Padua and Milan. Near the end of the section (now at page 11, lines 20-21, of the marked-up version) we wrote: "We can conclude that Trieste pressure of the 1841–1864 period should be considered suspect, …". In our opinion, that sentence contains what the reviewer asks for.**

Comment 6
20CR/20CRv3 might be a dynamically coherent product, but if it takes in your Trieste data before 1865 it is likely to be wrong. The differences are not that large, and it isn't formally looking for relatively small systematic anomalies. I suppose you are OK if you're not releasing the sub-daily data, but can you be sure of this into the future? In recent decades (since 1980), 20CRv3 will be taking in data from Milan, Padua and stations put into SYNOP messages by the Italian Met Service. This could include Trieste, but maybe a different site from yours.

**The Trieste data in SYNOP messages come from the station attended by the Met. Service of the Italian Air Force, which was close to ours in the past and was moved to a different site, several kilometres away, about 20 years ago.**
**We do not understand the other reviewer's remarks.**
**1) "… if it takes in your Trieste data before 1865 it is likely to be wrong". In Fig. 7a the black curve shows that TS-TS(20c) is coherent with TS-PD and TS-MI. If the 20CR curve was wrong because it ingested Trieste data, the black curve should have been much closer to zero. We rely on the fact that the data assimilation can detect too anomalous data, and gets rid of them or minimizes their impact.**
**2) "I suppose you are OK if you're not releasing the sub-daily data, but can you be sure of this into the future?". In fact, sub-daily data were not released. As for future developments, they will be considered for another work.**

**Answers to Report 2 (labelled as Report #1, Anonymous Referee #2) – Our replies in bold**

**As the reviewer's assessment is 'accepted as is', we thank him/her.**